# Systemic Therapy and Its Surgical Implications in Patients with Resectable Liver Colorectal Cancer Metastases. A Report from the Western Canadian Gastrointestinal Cancer Consensus Conference

Shahid Ahmed [1,*], Nicholas Bosma [2], Michael Moser [3], Shahida Ahmed [4], Bryan Brunet [1,5], Janine Davies [2], Corinne Doll [6], Dorie-Anna Dueck [1,7], Christina A. Kim [4], Shuying Ji [8], Duc Le [1,5], Richard Lee-Ying [6], Howard Lim [2], John Paul McGhie [9], Karen Mulder [10], Jason Park [11], Deepti Ravi [12], Daniel J. Renouf [2], Devin Schellenberg [13], Ralph P. W. Wong [4] and Adnan Zaidi [1,7,*]

1   Saskatoon Cancer Center, Saskatchewan Cancer Agency, 20 Campus Drive, University of Saskatchewan, Saskatoon, SK S7N 4H4, Canada; bryan.brunet@saskcancer.ca (B.B.); dorie-anna.dueck@saskcancer.ca (D.-A.D.); duc.le@saskcancer.ca (D.L.)
2   British Columbia Cancer Agency, Vancouver, BC V5Z 4E6, Canada; nicholas.bosma@bccancer.bc.ca (N.B.); jan.davies@bccancer.bc.ca (J.D.); hlim@bccancer.bc.ca (H.L.); drenouf@bccancer.bc.ca (D.J.R.)
3   Department of Surgery, College of Medicine, University of Saskatchewan, Saskatoon, SK S7N 5E5, Canada; mam305@mail.usask.ca
4   CancerCare Manitoba, Winnipeg, MB R3E 0V9, Canada; sahmed1@cancercare.mb.ca (S.A.); ckim3@cancercare.mb.ca (C.A.K.); rwong2@cancercare.mb.ca (R.P.W.W.)
5   Department of Radiation Oncology, College of Medicine, University of Saskatchewan, Saskatoon, SK S7N 5E5, Canada
6   Arnie Charbonneau Cancer Institute, Alberta Health Service, Calgary, AB T2N 4Z6, Canada; corinne.doll@albertahealthservices.ca (C.D.); richard.lee-ying@albertahealthservices.ca (R.L.-Y.)
7   Department of Medical Oncology, College of Medicine, University of Saskatchewan, Saskatoon, SK S7N 5E5, Canada
8   Shared Health, Winnipeg, MB R3B 2K6, Canada; sji@sharedhealthmb.ca
9   British Columbia Cancer Agency, Victoria, BC V8R 6V5, Canada; jmcghie@bccancer.bc.ca
10  Cross Cancer Institute, Alberta Health Services, Edmonton, AB T6G 1Z2, Canada; karen.mulder@albertahealthservices.ca
11  Department of Surgery, University of Manitoba, Winnipeg, MB R3T 2N2, Canada; jpark@manitoba-physicians.ca
12  Saskatchewan Health Authority, Saskatoon, SK S7K 0M7, Canada; deepti.ravi@saskhealthauthority.ca
13  British Columbia Cancer Agency, Surrey, BC V3V 1Z2, Canada; dschellenberg@bccancer.bc.ca
*   Correspondence: shahid.ahmed@saskcancer.ca (S.A.); adnan.zaidi@saskcancer.ca (A.Z.); Tel.: +306-655-2710 (S.A.); Fax: +306-655-0633 (S.A.)

**Abstract:** The Western Canadian Gastrointestinal Cancer Consensus Conference (WCGCCC) convened virtually on 4 November 2021. The WCGCCC is an interactive multi-disciplinary conference attended by health care professionals, including surgical, medical, and radiation oncologists; pathologists; radiologists; and allied health care professionals from across four Western Canadian provinces, British Columbia, Alberta, Saskatchewan, and Manitoba, who are involved in the care of patients with gastrointestinal cancer. They participated in presentation and discussion sessions for the purpose of developing recommendations on the role of systemic therapy and its optimal sequence in patients with resectable metastatic colorectal cancer.

**Keywords:** colorectal liver metastases; systemic therapy; chemotherapy; surgery; metastasectomy; adjuvant therapy; CRLM; consensus

## 1. Term of References

### 1.1. Purpose

The Western Canadian Gastrointestinal Cancer Consensus Conference (WCGCCC) aims to develop consensus opinions of oncologists and allied health professionals from

across Western Canada, attempting to define best care practices and to improve care and outcomes for patients with gastrointestinal cancers. The members participated in presentation and discussion sessions for the purpose of developing the recommendations on the role of systemic therapy and its optimal sequence in patients with resectable colorectal cancer liver metastases, both synchronous and metachronous metastatic disease. The recommendations are solely for resectable colorectal liver metastases and do not address borderline resectable or unresectable colorectal liver metastases.

### 1.2. Participants

The WCGCCC welcomes medical oncologists, radiation oncologists, surgical oncologists, pathologists, radiologists, gastroenterologists, and allied health professionals from western Canada who are involved in the care of patients with gastrointestinal malignancies (Table S1).

### 1.3. Target Audience

The recommendations presented here are targeted at health care professionals involved in the care of patients with colorectal cancers.

### 1.4. Basis of Recommendations

The recommendations are based on the presentation and discussion of the best available evidence. Where applicable, references are cited.

## 2. Consensus Questions

### 2.1. Q1: What Is the Role of Peri-Operative or Adjuvant Systemic Therapy in Patients with Resectable Metastatic Colorectal Cancer? *

- In patients with resectable colorectal liver metastases, the standard of care is upfront surgical resection.
- Adjuvant chemotherapy has been associated with better disease-free survival (DFS) in patients with resectable metastatic colorectal cancer and may be considered after resection, following a thorough discussion with the patient regarding the risks and benefits.
- Perioperative chemotherapy may be considered in selected patients where the surgical approach can be optimized after a multi-disciplinary team (MDT) discussion.
- A pre-operative chemotherapy duration of greater than 3 months is not recommended, given the risk of liver toxicity.
- There is no clear role for the use of biologics in this setting, outside of a clinical trial.

### 2.2. Q2: If Systemic Therapy Is a Consideration, What Is the Optimal Sequence of Systemic Therapy in Patients with Metastatic Colorectal Cancer?

- The optimal regimen and sequence of chemotherapy are not known.
- Oxaliplatin-based and single-agent fluoropyrimidine are both appropriate options.
- Sequencing of synchronous rectal cancers is more complex, with limited data, and requires multi-disciplinary discussion.

\* Note: Perioperative therapy is defined as 3 months of pre-operative or neoadjuvant and 3 months of post-operative or adjuvant combination chemotherapy. Adjuvant or postoperative treatment is defined as 6 months of single-agent or combination chemotherapy.

## 3. Introduction

Colorectal cancer is one of the most common cancers and is the leading cause of cancer-related death. The liver is the most common site of metastatic disease from colorectal cancer. Nearly 50% of patients with colon cancer will present with or develop liver metastasis [1]. Approximately 25% of colorectal liver metastases (CRLM) are deemed resectable, and surgery offers the best chance of a cure, improving 5-year survival rates by up to 40–50% [2–4]. However, disease recurrence is common, occurring in 60% of

patients or more [5,6]. Although adjuvant systemic therapy in early-stage colon cancer has demonstrated a reduction in the risk of relapse and increased long-term survival, the benefit of chemotherapy in the curative-intent surgical management of patients with resectable CRLM is less clear. The WCGCCC group reviewed the existing data and developed recommendations regarding the role of systemic therapy in patients with resectable CRLM.

## 4. Methods

The WCGCCC executive committee developed the consensus questions prior to the WCGCCC consensus meeting. Two members (SA and AZ) co-chaired the virtual consensus meeting. Two speakers (NB and MM) presented the evidence during a didactic session. Each provincial group subsequently reviewed the evidence and developed a consensus statement in response to consensus questions. The four provincial groups subsequently met, reviewed all four-consensus statements, and discussed various aspects of evidence. The participants reached a final consensus statement, agreed by the four provincial groups.

## 5. Results: Summary of the Evidence

### 5.1. Surgical Management of CRLM

There have been considerable advances in the number of surgical options available for the management of CRLM over the past two decades. As recently as the early 1990s, the management of a patient with CRLM was largely palliative, and liver resection in these cases was considered controversial [7].

### 5.1.1. Evolving Definition of Resectability

In addition to the acceptance that liver resection for colorectal metastases could offer patients improved survival, the redefinition of resectability was also a significant advance. In the late 1990s, criteria were established for resectability that were based on the number and size of metastases and the need to have a margin of 1 cm or more [7]. More recently, the definition of resectability has evolved to include resection of all liver metastases, as long as there is enough residual functional liver for survival. Specifically, a microscopically negative margin is sufficient, although recent evidence points to superior results with a margin greater than 1 mm [8]. There must be preservation of two contiguous functional liver segments with an intact portal and arterial inflow, venous outflow, and biliary drainage. Finally, prior to embarking upon aggressive resections, there must also be an adequate future liver remnant (FLR) [9]. In cases where the FLR is felt to be insufficient, maneuvers, such as portal vein embolization or ligation, can be employed to increase the FLR prior to resection.

### 5.1.2. Newer Options for Surgical Resection of CRLM

There are many new options available for surgical or ablative clearance of the liver of metastases. In any given case, there is often more than one possible option, with each having its own advantages and disadvantages for discussion with the patient and in multi-disciplinary tumor board rounds.

### 5.1.3. Ablation and Two-Stage Procedures

Ablation techniques, such as radiofrequency ablation and microwave ablation, have been available since the 1990s and were employed in a patient with one or two small metastases in the lobe contralateral to that being resected [10]. This led to a modest increase in the resectability of CRLM. Two-stage hepatectomy further improved resectability rates and was first reported in some large series around the year 2000 [1,7]. This strategy included surgical resection, followed by a period of one to six months to allow for regeneration to occur in the remaining liver before planning for the clearance of the remaining tumor in a second operation [11]. However, regeneration was gradual, and in the time between operations, some patients experienced disease progression.

### 5.1.4. Portal Vein Embolization and Ligation

Around the same time, portal vein embolization, performed percutaneously, was introduced as a way of preparing for resection. The portal vein on the side planned for resection was embolized with a variety of materials. Follow-up imaging could gauge the response in terms of atrophy of that lobe and hypertrophy of the contralateral lobe. This was used in cases where the FLR was felt to be borderline or inadequate, as in some cases of tri-segmental resection [12].

The combination of two-stage hepatectomy along with portal vein embolization or ligation brought together the advantages of both approaches. Such a strategy made further significant gains in the resectability of CRLM, with one institution reporting an increase in resectability from about 40% to about 70% of all patients referred for CRLM [13].

### 5.1.5. Associating Liver Partition and Portal Vein Ligation for Staged Hepatectomy (ALPPS)

Associating Liver Partition and Portal vein ligation for Staged hepatectomy (ALPPS) is a strategy that was first attempted somewhat by accident when a planned two-stage procedure for a patient with bilobar colorectal liver metastases had to be aborted [14]. The parenchyma was transected and the portal vein was ligated, but the lobe that was planned to be removed was not resected. A CT scan performed one week later showed an incredible hypertrophy of the remaining left lateral segment, to almost double in size, along with significant atrophy of the right lobe. This was much more than would have been expected from simple portal vein ligation, and this accelerated regeneration over the span of one week is thought to be related to the transection and the raw surface that resulted.

The introduction of ALPPS further increased the apparent resectability of CRLM to 80% or 85% [7]. Unfortunately, the initial series of ALPPS procedures demonstrated a high rate of complications, including rates of in-hospital mortality in some series in the 20% to 29% range [7,15]. The rate of morbidity was reported around 50% to 80% [15]. There were also high rates of early recurrence. A more recent experience with ALPPS has fortunately shown reduced morbidity and mortality, as well as decreased recurrence rates [7,15].

### 5.1.6. Liver Transplantation Options

Finally, in some large centers, using very specific and strict criteria, liver transplantation is again being assessed as a possible treatment strategy for CRLM [16]. Other strategies taking advantage of regeneration of a small liver segment while the affected liver is still in situ, known as RAPID (i.e., resection and partial liver segment 2–3 transplantation with delayed total hepatectomy) have also been described [7].

Over the span of two decades, the resectability of colorectal metastases essentially went from 0% to 20%, to 70%, and now up to 90% in patients without medical contraindications [7]. This rapid increase in surgical options for the management of CRLM coincided with the introduction of new chemotherapy regimens.

### 5.2. Surgical Implications of Systemic Therapy for Colorectal Liver Metastases

Nearly a full decade after the introduction of oxaliplatin-based chemotherapy for CRLM, a group of prominent CRLM surgeons and researchers penned an editorial, a cautionary note, detailing some of the effects of neoadjuvant chemotherapy that had been observed in patients undergoing liver resections [17].

Around that same time, a report on the histological findings in 174 CRLM liver resection specimens was published showing sinusoidal obstruction of the liver surrounding the resected liver metastasis in 79% of patients treated with oxaliplatin. They also documented nodular regenerative hyperplasia in 16% and steatosis in 49% [18].

### 5.2.1. Sinusoidal Obstruction Syndrome (SOS)

SOS is a condition previously observed in the form of venoocclusive disease (VOD), noted in patients undergoing chemotherapy prior to stem cell transplant. Microscopically,

there is an obvious injury to the endothelial lining of the sinusoids, thrombosis, and extravasation of red blood cells. Grossly, the congestion that results from obstruction of flow proceeding to the central vein gives the liver a mottled and bluish appearance.

Vauthey confirmed the association of oxaliplatin with SOS [19], and despite the dramatic bluish appearance of the liver in these cases, his group found no increase in 90-day morbidity or mortality in 79 patients treated with oxaliplatin who had undergone resection.

Nakano et al. reported in 2008 that about 52% of patients treated with oxaliplatin in their series developed SOS [20]. They did note some increased morbidity in terms of grade 3 and grade 4 Clavien-Dindo complications (6.3% vs. 40%, *p* = 0.026), and an increased hospital stay, although both the sinusoidal-injury-positive and sinusoidal-injury-negative groups were admittedly small, with fewer than 20 patients in each group.

The EORTC 40983 trial [21] was a multicenter, randomized controlled trial comparing surgery only to oxaliplatin-based chemotherapy before and after surgery for upfront resectable CRLM. As in Vauthey's study, they noted a low rate of complications in the group receiving oxaliplatin-based chemotherapy prior to resection.

Soubrane, in 2009 published the results of a retrospective look at patients who had undergone major hepatectomy following pre-operative chemotherapy [22]. Among those treated with oxaliplatin, 59% of these patients developed SOS. There was a trend towards increased blood loss and an increased volume of transfusion in the group with higher grade SOS, compared to the group with lower grade SOS. They did, however, see an increase in transient hepatic dysfunction (3/13 vs. 26/38, *p* = 0.004) and ascites as well. The only cases of severe liver failure were seen in patients with high-grade SOS lesions.

Overall, in the two largest series, it would not appear that SOS is of great consequence to liver resection, although one must remember that most of the studies involved rather small groups, were retrospective, and were of course performed in high-volume centers, which may have a lower rate of complications in general.

### 5.2.2. Nodular Regenerative Hyperplasia (NRH)

NRH has some similarities and common associations with SOS. Grossly, a liver affected by NRH can look almost cirrhotic, and yet microscopically, it is notable for its nodularity in the absence of significant fibrosis. NRH is associated with sinusoidal dilatation and is also associated with oxaliplatin treatment. Vigano in 2015 reported on 406 patients with colorectal liver metastases undergoing resection after chemotherapy and identified 87 (18%) patients with NRH. NRH was found to be associated with an increased risk of transient liver failure (9% vs. 2%, *p* = 0.002), with a higher risk associated with major resection [23].

### 5.2.3. Steatohepatitis and Steatosis

Steatohepatitis was documented in 20.2% of patients receiving irinotecan in a study of 248 patients receiving pre-operative chemotherapy followed by liver surgery for CRLM [19]. The same group also noted an increased risk of steatohepatitis in patients with a higher body mass index (BMI). Patients in this series with steatohepatitis had a higher 90-day mortality post-resection than those without steatohepatitis (14.7% vs. 1.6%, OR = 10.5, *p* = 0.001). A case-control study of 102 patients with steatohepatitis matched to controls without steatohepatitis but with a similar extent of resection noted similar results [24]. Those with steatohepatitis had increased overall morbidity and a higher risk of hepatic decompensation.

Whether steatosis without the inflammatory cell infiltrates of steatohepatitis leads to an increased risk of morbidity or mortality has been controversial. However, a well-constructed meta-analysis suggested that the risk of complications is approximately doubled in patients with significant steatosis, whereas there was no obvious increase in mortality following liver resection compared with those patients without steatosis [25].

### 5.2.4. Effects of Bevacizumab

Bevacizumab is a monoclonal antibody often used in the treatment of patients with metastatic colorectal cancer. Increased risk of infection and poor wound healing are potential complications of bevacizumab [26]. Waiting for five to eight weeks after the last dose of bevacizumab before proceeding with surgical intervention is advised and has reduced the incidence of these wound complications [26,27]. As an added benefit, bevacizumab seems to have a protective effect against SOS. This was documented in a histopathological study of patients treated with oxaliplatin, with and without bevacizumab [27]. The group documented a significant reduction in the incidence of grades 2 or 3 SOS when bevacizumab was added (62% vs. 31%, $p < 0.001$).

### 5.2.5. Limitations to the Data Surrounding CRLM

There are several limitations to the data surrounding surgical complications related to pre-operative chemotherapy. Firstly, as in any surgical research study, there will always be a significant amount of surgical variation between centers and between surgeons at those centers. There is no consensus as to what constitutes resectable and unresectable disease, and any given case can have different surgical approaches in terms of the transection device, the philosophy of major liver resection vs. multiple wedge resections, and the extent to which a given institution uses PVE or PVL [28]. Almost all the studies were retrospective and observational, and some drew conclusions based on relatively small numbers of patients.

Finally, one must recall that these reports generally came from high-volume centers. Although no increase in morbidity or mortality was reported for several of the post-chemotherapy pathologies, it is quite possible that these differences may be more apparent in a small-volume center.

### 5.2.6. Should Hepatic Injury Change Surgical Planning?

Steatohepatitis, in particular, seems to increase the morbidity and mortality of liver resection, whereas it is less clear whether SOS, in spite of the dramatic mottled blue appearance of the liver, has any effect on surgical morbidity and mortality. However, preoperative imaging and even needle biopsy are not particularly helpful. While imaging might identify steatosis in up to 50% of patients, it cannot distinguish the more serious steatohepatitis from steatosis. Likewise, the irregular distribution of the disease process in SOS makes it such that a needle biopsy can underrepresent the extent of disease elsewhere in the liver. The intraoperative appearance can help to make the diagnosis, but does not give a full picture. While the intraoperative finding of steatosis may not cause most surgeons to abandon the procedure, these appearances might make one think very carefully about reducing the extent of liver tissue that is removed. Specifically, one might consider multiple wedge resections and ablation as opposed to a formal lobectomy in a very steatotic liver.

### 5.2.7. What Is the Ideal Number of Cycles of Chemotherapy before Resection?

Because of the changing and continually evolving chemotherapy agents and regimens, the data surrounding an ideal number of treatments prior to surgery has been quite heterogeneous. For cases where the goal is conversion therapy (the conversion of an unresectable situation into a resectable one), the number of chemotherapy treatments will be dictated by the response and either conversion to a resectable situation or disease progression. In those cases where neoadjuvant chemotherapy is contemplated in cases of resectable disease, a few studies have consistently suggested that six or fewer cycles might be optimal, with morbidity increasing when more than six cycles are utilized [29,30].

### 5.2.8. How Long after Chemotherapy Should Liver Resection Take Place?

Likewise, there is not a lot of evidence in the literature as to how long one should wait after the final dose of chemotherapy before proceeding to liver resection. This is a balance between greater toxicity to the liver if the operation takes place too soon after the last dose

of chemotherapy vs. the risk of disease progression if the interval between chemotherapy and surgery is too long. The data are also quite heterogeneous, and yet several studies, generally retrospective in nature, some with small groups, consistently suggest an interval between five and eight weeks [31].

5.2.9. Reduction of the Extent of an Aggressive Surgical Resection and Assessment of Biological Behavior

Although the oncologic data suggest improved disease-free survival but no obvious overall survival advantage to pre-operative and post-operative chemotherapy for CRLM, there are some surgical reasons to consider neoadjuvant chemotherapy. With aggressive strategies, including two-stage procedures with or without portal vein embolization/ligation, or ALPPS, the majority of patients with CRLM can be considered resectable; however, this will be at the cost of a major surgery and its attendant risks. In these cases, neoadjuvant chemotherapy might make it possible to reduce the extent/severity of the surgery while at the same time acting as a biological test of time. The goal would be to assess the biological behavior of the tumor. Conversely, a tumor that is seen as having more favorable biological behavior during the test of time of chemotherapy, on some level, helps to justify a major resection. This test of time may be one of the reasons attributed to the improvement of ALPPS results over the last five years [32].

*5.3. Long-Term Dynamic Changes of NMDA Receptors following an Excitotoxic Challenge*
5.3.1. Benefit of Systemic Therapy in Resectable Metastatic Liver Disease

Several randomized controlled trials (RCTs) have explored various peri-operative chemotherapy regimens in the management of resectable CRLM [33,34]. In patients with upfront resectable CRLM, doublet regimens, such as FOLFIRI (infusional 5FU/leucovorin and irinotecan) [35] or FOLFOX (infusional 5FU/leucovorin and oxaliplatin) [21,36,37], or single-agent fluoropyrimidines [38,39] have consistently demonstrated clinically significant improvements in DFS, but this has not translated to an overall survival (OS) benefit [35–39]. Furthermore, systematic reviews and meta-analyses have attempted to clarify the benefit of curative-intent peri-operative systemic therapy for CRLM [40–49]. However, the lack of prospective RCTs and inclusion of several observational studies [49,50] has limited the interpretation of the guidance of clinical practice [40,42]. In addition, several of these studies include regional non-systemic approaches, such as hepatic-arterial infusion (HAI) [47,49,51] that are not widely available or utilized.

A total of five RCTs have compared the addition of peri-operative chemotherapy with resection of CRLM vs. surgery alone, dating back 20 years [36,37,39,50,52] [Table 1]. All the trials were powered with the primary endpoint of DFS, with the exception of Langer et al. [50], which was ultimately closed prematurely due to poor accrual. Three of the trials examined adjuvant monotherapy fluoropyrimidine [39,50,52]; however, Hasegawa et al. [52] used uracil-tegafur, which is not standard practice in North America. The pooled analysis of FFDC and ENG trials that used 5FU/leucovorin monotherapy showed that adjuvant chemotherapy was associated with a median progression-free survival (PFS) of 27.9 months compared to 18.8 with observation alone ($p = 0.058$). Likewise, the group who received adjuvant chemotherapy had a median OS of 62.2 months compared to 47.3 months with observation alone ($p = 0.095$) [38]. The EORTC 40983 trial by Nordlinger et al. examined the role of peri-operative (three months pre- and post-operative) mFOLFOX4 [36]. The most recently published trial by Kanemitsu et al. (JCOG0603) in 2021 investigated adjuvant mFOLFOX6 after hepatectomy [37]. The five-year DFS in these trials was improved from about 30% to 40–50% with an approximate 30% reduction in recurrence or death. All the RCTs met the primary endpoint of DFS, but none were able to capture a statistically significant benefit in the underpowered secondary endpoint of OS. This was recently corroborated in a meta-analysis with a statistically significant improvement in DFS (HR 0.71, 95% CI: 0.61–0.82; $p < 0.001$) but not OS (HR 0.87, 95% CI: 0.73–1.04; $p = 0.136$) with the use of adjuvant chemotherapy [53].

**Table 1.** Summary of randomized clinical trials that assessed peri-operative or adjuvant chemotherapy in patients with metastatic colon cancer with resectable liver disease.

| Trial | Number | Interventions | DFS | OS |
|---|---|---|---|---|
| Peri-operative chemotherapy | | | | |
| EORTC 40983 [21,36] | 364 | Peri-operative FOLFOX for 12 cycles vs. observation | 3-year DFS rate 36.2 vs. 28.1% (HR 0.77, $p = 0.041$ | Median OS 63.7 vs. 55 months (HR 0.84, $p = 0.3$) |
| Adjuvant chemotherapy | | | | |
| ENG Trial [50] | 129 | 5FU/LV vs. observation | 4-years DFS rate 45 vs. 35% ($p = 0.35$) | 4-year OS rate 57 vs. 47% ($p = 0.39$) |
| FFCD trial [39] | 173 | 5FU/LV vs. observation | 5-year DFS 33.5 vs. 26.7% ($p = 0.028$) | 5-year OS rate 51.1 vs. 41.1% ($p = 0.13$) |
| Ychou et al. [35] | 321 | FLOFIRI vs. 5FU/LV | Median PFS 24.7 vs. 21.6 months | 3-year OS rate 72.7 vs. 71.6% ($p = 0.028$) |
| Hasegawa et al. [52] | 180 | Uracil-tegafur/LV vs. observation | 3-year RFS 38.6 vs. 32.3% (HR 0.56, $p = 0.003$) | 5-year OS 66.1 vs. 66.8%, HR 0.8, $p = 0.4$) |
| JCOG603 [37] | 300 | FOLFOX vs. observation | 3-year DFS 52.1 vs. 41.5% (HR 0.63, $p = 0.002$) | 5-year OS rate 69.5 vs. 83% |

DFS: Disease-free survival; HR: hazard ratio; RFS: relapse-free survival; OS: overall survival.

The most widely cited randomized trial in this setting is the EORTC 40983 trial by Nordlinger et al. [21,36]. This trial demonstrated a DFS benefit, but only after a post hoc sensitivity analysis of "eligible patients." Regardless, this did not translate to an improvement in OS [36]. The authors' comparisons were between the statistically non-significant 4.1% absolute OS benefit at five years in their study and that of the statistically significant 4.2% six-year OS absolute benefit seen with the use of adjuvant FOLFOX4 after resection of early-stage colon cancer in the MOSAIC trial [34]. A possible explanation for the absence of an OS benefit is the smaller sample sizes and insufficient power to detect differences in OS as a primary endpoint in trials of CRLM.

The JCOG0603 trial demonstrated a significant absolute improvement in DFS of 2.6 years (4.3 vs. 1.7 years) with adjuvant mFOLFOX6; however, there was a numerical trend to the worse five-year OS of 71.2% vs. 83.1% [37]. The authors observed an unanticipated poor tolerance to mFOLFOX6 in the initial phase 2 portion of the study, with only 36% of patients completing all planned cycles of chemotherapy due to severe adverse effects. The authors argued that this may have led to a detrimental effect on the observed OS. Although there was no statistical analysis, the authors presented a significantly improved appearance of Kaplan–Meier OS curves from the latter phase 2/3 data, where an amendment was made to the treatment protocol that resulted in an improvement to approximately 70% compliance and completion of the adjuvant chemotherapy cycles. Similarly, the post-operative component of mFOLFOX4 in the EORTC trial also demonstrated lower completion rates of 47% [21]. Poor tolerance to post-operative oxaliplatin-based chemotherapy may be due to specific toxicities, such as oxaliplatin-induced hepatic SOS in the context of compromised liver parenchyma following resection of CRLM [54]. Other possible reasons for poor tolerance to oxaliplatin-based chemotherapy may be cumulative toxicities, such as neuropathy, post-operative complications, slow recovery, and worse overall performance status. Poor compliance and low completion rates with oxaliplatin-based regimens could, at least in part, explain the differences seen in survival outcomes. There was evidence of improved tolerance to pre-operative mFOLFOX4 in the EORTC trial with approximately 80% of patients completing the chemotherapy cycles. The ongoing CHARISMA trial investigating neoadjuvant CAPOX chemotherapy in resectable but high-risk CRLM, compared with surgery alone, may help to clarify the optimal sequencing

strategy of peri-operative chemotherapy [55]. Pre-operative chemotherapy also has the added benefits of assessing tumor biology, downstaging disease, and improving surgical resection, and may also adequately address micrometastatic disease [56].

### 5.3.2. The Optimal Systemic Therapy in Resectable CRLM

If peri-operative chemotherapy is offered in resectable CRLM, the optimal regimen is also unclear. There is currently no evidence to suggest the use of biologics, and they may in fact be more detrimental. For example, worse survival outcomes were observed in the new EPOC trial with the addition of cetuximab to chemotherapy [57]. A recent meta-analysis stratifying monotherapy fluoropyrimidine vs. oxaliplatin-based chemotherapy showed a non-statistical trend favoring the former (HR 0.76, 95% CI: 0.57–1.01; $p = 0.059$) [53]. It is possible that this relates to poor tolerance and complications of post-hepatectomy oxaliplatin chemotherapy. There have been no trials examining irinotecan-based regimens to surgery alone in resectable CRLM, but one trial showed no differences between 5FU and FOLFIRI [35].

Current guidelines for resected CRLM lack consensus and generally follow practices for the management of adjuvant early-stage colon cancer, in favor of an oxaliplatin-based regimen in the peri-operative setting [58]. With the recent publication of JCOG0603, there is mounting evidence to suggest a consistent improvement in DFS, but not necessarily OS, for the addition of chemotherapy to resected CRLM. However, there is a paucity of adequately powered RCTs as well as significant clinical heterogeneity within the relevant studies that result in difficult interpretations. Despite these limitations, the current evidence suggests that clinicians should reform the framework for discussion with patients instead of reflexively prescribing peri-operative chemotherapy for resected CRLM [59]. These conversations will continue to evolve and become increasingly complex with greater involvement of the patient in decision-making, outlining the risks and benefits of such an approach. It may still be reasonable and appropriate to consider the addition of chemotherapy to resection of CRLM in patients motivated by prolonging recurrence but not necessarily long-term survival.

## 6. Conclusions

In summary, upfront surgery is the preferred approach in patients with resectable CRLM. Adjuvant chemotherapy in patients with resected CRLM has been associated with a significant reduction in the risk of recurrent colon cancer. Oxaliplatin-based and single-agent fluoropyrimidine are both appropriate options. If there is a consideration of preoperative chemotherapy, a duration of greater than 3 months is not recommended, given the risk of liver toxicity. Both adjuvant and peri-operative chemotherapy have not demonstrated an improvement in overall survival.

**Supplementary Materials:** The following are available online at https://www.mdpi.com/article/10.3390/curroncol29030147/s1, Table S1: List of participants from the Western Canadian Gastrointestinal Cancers Group.

**Author Contributions:** S.A. (Shahid Ahmed), N.B., M.M. and A.Z. wrote the initial draft. S.A. (Shahid Ahmed), N.B., M.M., S.A. (Shahida Ahmed), B.B., J.D., C.D., D.-A.D., C.A.K., S.J., D.L., R.L.-Y., H.L., J.P.M., K.M., J.P., D.R., D.J.R., D.S., R.P.W.W. and A.Z. were involved in the development of consensus statement, reviewing the draft and making appropriate changes. All authors have read and agreed to the published version of the manuscript.

**Funding:** The WCGCCC received funding from Pfizer Canada Inc., Eisai Inc., IPSEN Biopharmaceutical Canada Inc., Viatris, Taiho Pharma Canada Inc., Incyte Biosciences Canada, Bristol-Myers Squibb Canada, AstraZeneca, and Amgen Canada Inc. However, no sponsor was involved in the development of consensus questions and the program, discussion and consensus statement, and writing or reviewing the manuscript.

**Institutional Review Board Statement:** Not applicable.

**Informed Consent Statement:** Not applicable.

**Data Availability Statement:** Not applicable.

**Acknowledgments:** We would like to extend our gratification to the meeting participants for their contribution to developing consensus statements, and Shenai Recile and Maureen Melnyk for their administrative support.

**Conflicts of Interest:** Author Shahid Ahmed has served on advisory boards for Merck, BMS, Pfizer, Taiho and Roche. Author Karen Mulder has an advisory role for Pfizer Canada, Eisai Inc., Bayer Canada and has received clinical trial funding from Deciphera Pharmaceuticals, BluePrint Medicines, and AstraZeneca. Author Janine Davies has clinical trials for BMS, Merck, MedImmune, Astellas Array BioPharma, and is a consultant and a member of the advisory board for AstraZeneca, Eisai, Taiho, and Amgen. Author Christina A. Kim received an unrelated research grant from Celgene Inc. and an honorarium from Amgen. Author Howard Lim received honoraria from Merck, BMS, AstraZeneca, Eisai, Taiho, Roche, Amgen, and Bayer for consultant work. Author Richard Lee-Ying had advisory roles for Eisai, Ipsen, AstraZeneca, Roche, and Celgene. Author Daniel J. Renouf received unrelated research funding and honoraria from Bayer and Roche, and travel funding and honoraria from Servier, Celgene, Taiho, Ipsen, and AstraZeneca. Author Adnan Zaidi has provided consultant work for Merck and BMS. The remaining authors declare no conflict of interest.

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
