# Peer review of "Systemic Therapy and Its Surgical Implications in Patients with Resectable Liver Colorectal Cancer Metastases. A Report from the Western Canadian Gastrointestinal Cancer Consensus Conference"

_curroncol, doi:10.3390/curroncol29030147_

Round 1
Reviewer 1 Report
The role for peri-operative chemotherapy in the setting of resectable metastatic CRC remains uncertain in the face of evidence to date. Thus consensus guidelines and statements such as this manuscript take on particular importance to inform current clinical practice.
Recommendations:
- The title is not clear relative to manuscript content; while resectable metastatic colorectal cancer can include lung and other extra-hepatic sites, the reviewed evidence is restricted to liver metastatic disease. Please modify.
- The title does not give credit to the fulsome discussion that has been included on surgical management in Section 3.1 (which I found to be excellent, by the way)
- I suggest for clarity that the authors specifically state, e.g. in the Terms of Reference section, that the manuscript does not cover cases of potentially resectable disease where downstaging/sizing is intended
- While Table 1 informs reader about diversity of multidisciplinary group, the manuscript does not need it in main paper
- Section 2.1 Q1 - to add clarity for reader, define to some extent what is meant by peri-op and adjuvant, e.g. 12 cycles 6 pre- or 12 post- etc.; I would prefer term post-operative to 'adjvuant' in this question
- Section 2.2 Q1 - bullet 1 - please clarify whether consensus includes the upfront surgical resection of primary in synchronous cases (versus metachronous)
- Section 2.2 Q1 - bullet 2 - incomplete, as the reader will be biased by mention of improved DFS while authors have excluded discussion around lack of corresponding OS advantage
- Section 2.2 Q2 - please clarify question; it is not clear to reader if the term 'sequence' refers to pre- versus peri- versus post-operative chemotherapy; I'm not clear why second bullet point is included in this question about sequence; if including third bullet point, clarify that you are (presumably) referring to the placement of chemo-radiation in managing rectal cancer
- Section 3.3.1 is complete with respect to references included and balanced presentation of the evidence
- Section 3.3.2 is also well written and does not require edits
Author Response
- The title is not clear relative to manuscript content; while resectable metastatic colorectal cancer can include lung and other extra-hepatic sites, the reviewed evidence is restricted to liver metastatic disease. Please modify.
Thanks very much for this comments. As per suggestions, we have modified the title.
“Systemic therapy and its implications in surgical management of patients with resectable colorectal liver metastases. A report from the Western Canadian Gastrointestinal Cancer Consensus Conference”
- The title does not give credit to the fulsome discussion that has been included on surgical management in Section 3.1 (which I found to be excellent, by the way)
Thanks very much. As per suggestions, we have modified the title and incorporated it in the revised title.
- I suggest for clarity that the authors specifically state, e.g. in the Terms of Reference section, that the manuscript does not cover cases of potentially resectable disease where downstaging/sizing is intended
Thanks very much for this comment. We have specified it in the revised paper (Section 1.1) that the recommendations are solely for resectable colorectal liver metastases and do not address borderline resectable or unresectable colorectal liver metastases.
“The Western Canadian Gastrointestinal Cancer Consensus Conference (WCGCCC) aims to develop ……………………………………resectable or unresectable colorectal liver metastases”.
- While Table 1 informs reader about diversity of multidisciplinary group, the manuscript does not need it in main paper
Thank you for the suggestion. We have removed table 1 from the main text and have added as a supplemental table (Table S1)
- Section 2.1 Q1 - to add clarity for reader, define to some extent what is meant by peri-op and adjuvant, e.g. 12 cycles 6 pre- or 12 post- etc.; I would prefer term post-operative to 'adjvuant' in this question
Thanks very much for the comments. Since the consensus questions and the recommendations were developed by the entire WCGCCC group we are not able to make changes in section 2. However, in the section 2 with a footnote we have clarified the duration of perioperative and adjuvant or post-operative therapy.
“Perioperative therapy is defined as 3 months of pre-operative or neoadjuvant and 3 months of post-operative or adjuvant combination chemotherapy. Adjuvant or post-operative treatment is defined as 6 months of single agent or combination chemotherapy”.
- Section 2.2 Q1 - bullet 1 - please clarify whether consensus includes the upfront surgical resection of primary in synchronous cases (versus metachronous)
It has been clarified in the revised paper in section 1.1 that the consensus included both synchronous and metachronous colorectal liver metastases.
- Section 2.2 Q1 - bullet 2 - incomplete, as the reader will be biased by mention of improved DFS while authors have excluded discussion around lack of corresponding OS advantage
Thanks very much for the comments. Since the consensus questions and the recommendations developed by the WCGCCC group we are not able to make changes in section 2. However, it is clarified in page 17 of section 5.3.1 that “ all the RCTs met the primary endpoint of DFS, but none were able to capture a statistically significant benefit in the underpowered secondary endpoint of OS”.
- Section 2.2 Q2 - please clarify question; it is not clear to reader if the term 'sequence' refers to pre- versus peri- versus post-operative chemotherapy; I'm not clear why second bullet point is included in this question about sequence; if including third bullet point, clarify that you are (presumably) referring to the placement of chemo-radiation in managing rectal cancer
Thanks very much for the comments. Since the consensus questions and the recommendations developed by the WCGCCC group we are not able to make changes in section 2. The sequencing included multiple scenarios in both rectal and colon cancers and involved neoadjuvant chemotherapy (all treatment before surgery), adjuvant (all treatment after surgery) or perioperative treatment. With respect to bullet 2 the WCGCCC group felt that it is important to specify the preferred regimen.
- Section 3.3.1 is complete with respect to references included and balanced presentation of the evidence
Thanks very much for the comments.
- Section 3.3.2 is also well written and does not require edits
Thanks very much for the comments.
- The title is not clear relative to manuscript content; while resectable metastatic colorectal cancer can include lung and other extra-hepatic sites, the reviewed evidence is restricted to liver metastatic disease. Please modify.
Thanks very much for this comments. As per suggestions, we have modified the title.
“Systemic therapy and its implications in surgical management of patients with resectable colorectal liver metastases. A report from the Western Canadian Gastrointestinal Cancer Consensus Conference”
- The title does not give credit to the fulsome discussion that has been included on surgical management in Section 3.1 (which I found to be excellent, by the way)
Thanks very much. As per suggestions, we have modified the title and incorporated it in the revised title.
- I suggest for clarity that the authors specifically state, e.g. in the Terms of Reference section, that the manuscript does not cover cases of potentially resectable disease where downstaging/sizing is intended
Thanks very much for this comment. We have specified it in the revised paper (Section 1.1) that the recommendations are solely for resectable colorectal liver metastases and do not address borderline resectable or unresectable colorectal liver metastases.
“The Western Canadian Gastrointestinal Cancer Consensus Conference (WCGCCC) aims to develop ……………………………………resectable or unresectable colorectal liver metastases”.
- While Table 1 informs reader about diversity of multidisciplinary group, the manuscript does not need it in main paper
Thank you for the suggestion. We have removed table 1 from the main text and have added as a supplemental table (Table S1)
- Section 2.1 Q1 - to add clarity for reader, define to some extent what is meant by peri-op and adjuvant, e.g. 12 cycles 6 pre- or 12 post- etc.; I would prefer term post-operative to 'adjvuant' in this question
Thanks very much for the comments. Since the consensus questions and the recommendations were developed by the entire WCGCCC group we are not able to make changes in section 2. However, in the section 2 with a footnote we have clarified the duration of perioperative and adjuvant or post-operative therapy.
“Perioperative therapy is defined as 3 months of pre-operative or neoadjuvant and 3 months of post-operative or adjuvant combination chemotherapy. Adjuvant or post-operative treatment is defined as 6 months of single agent or combination chemotherapy”.
- Section 2.2 Q1 - bullet 1 - please clarify whether consensus includes the upfront surgical resection of primary in synchronous cases (versus metachronous)
It has been clarified in the revised paper in section 1.1 that the consensus included both synchronous and metachronous colorectal liver metastases.
- Section 2.2 Q1 - bullet 2 - incomplete, as the reader will be biased by mention of improved DFS while authors have excluded discussion around lack of corresponding OS advantage
Thanks very much for the comments. Since the consensus questions and the recommendations developed by the WCGCCC group we are not able to make changes in section 2. However, it is clarified in page 17 of section 5.3.1 that “ all the RCTs met the primary endpoint of DFS, but none were able to capture a statistically significant benefit in the underpowered secondary endpoint of OS”.
- Section 2.2 Q2 - please clarify question; it is not clear to reader if the term 'sequence' refers to pre- versus peri- versus post-operative chemotherapy; I'm not clear why second bullet point is included in this question about sequence; if including third bullet point, clarify that you are (presumably) referring to the placement of chemo-radiation in managing rectal cancer
Thanks very much for the comments. Since the consensus questions and the recommendations developed by the WCGCCC group we are not able to make changes in section 2. The sequencing included multiple scenarios in both rectal and colon cancers and involved neoadjuvant chemotherapy (all treatment before surgery), adjuvant (all treatment after surgery) or perioperative treatment. With respect to bullet 2 the WCGCCC group felt that it is important to specify the preferred regimen.
- Section 3.3.1 is complete with respect to references included and balanced presentation of the evidence
Thanks very much for the comments.
- Section 3.3.2 is also well written and does not require edits
Thanks very much for the comments.
Reviewer 2 Report
Dear authors, your manuscript is very interesting but major flaws exist. Therefore, your manuscript cannot yet be accepted for publication:
1. There should be a small introduction to the subject. Why is such a study necessary? Gives us the background for your study in the purpose section.
2. Table 1 should be removed from the main manuscript and participant names should be provided as additional material or annex.
3. The structure of your paper is confusing. Provide a methods section to clarify all this and guide the reader for what he is about to read. Restructure your paper to make it easy and pleasant to read.
4. You provide the consensus questions with statements, where are the results of the answers? The answers to these statements must be easily found by readers in your manuscript.
5. What are the limitations of your study?
6. What are the conclusions of your consensus? Please provide a paragraph at the end of your manuscript.
Author Response
- There should be a small introduction to the subject. Why is such a study necessary? Gives us the background for your study in the purpose section.
Thanks very much for the comments. We have added a small introduction (Section 3) to clarify the background.
“Colorectal cancer is one of the most common cancers and is the leading cause of cancer-related death…………………………………………….in patients with resectable CRLM”.
- Table 1 should be removed from the main manuscript and participant names should be provided as additional material or annex.
As per suggestion we have removed table 1 from the main text and have added as a supplement table (Table S1)
- The structure of your paper is confusing. Provide a methods section to clarify all this and guide the reader for what he is about to read. Restructure your paper to make it easy and pleasant to read.
Thanks very much for the suggestion. As per suggestion we have restructured the paper and have added introduction (section 3), methods (section 4), retitled evidence to results (section 5) and added a brief conclusions (Section 6).
Methods: “The WCGCCC executive committee …………………….. reached to a final consensus statement agreed by the four provincial groups”.
- You provide the consensus questions with statements, where are the results of the answers? The answers to these statements must be easily found by readers in your manuscript.
Thanks very much for the comments. The statements are in response to the consensus questions. The evidence are provided in the result section (Section 5 in the revised paper).
- What are the limitations of your study?
This was a consensus guideline and limitations of current evidence were discussed in the result section.
- What are the conclusions of your consensus? Please provide a paragraph at the end of your manuscript.
A conclusions section (section 6) has been added in revised paper.
Conclusions: In summary upfront surgery is the preferred approach in patients …………………………………………..perioperative chemotherapy have not demonstrated improve in overall survival.